# Fatty Liver as Potential Biomarker of Atherosclerotic Damage in Familial Combined Hyperlipidemia

**DOI:** 10.3390/biomedicines10081770

**Published:** 2022-07-22

**Authors:** Giuseppe Mandraffino, Carmela Morace, Maria Stella Franzè, Veronica Nassisi, Davide Sinicropi, Maria Cinquegrani, Carlo Saitta, Riccardo Scoglio, Sebastiano Marino, Alessandra Belvedere, Valentina Cairo, Alberto Lo Gullo, Michele Scuruchi, Giovanni Raimondo, Giovanni Squadrito

**Affiliations:** 1Lipid Center, Department of Clinical and Experimental Medicine, University of Messina, 98122 Messina, Italy; cmorace@unime.it (C.M.); mscuruchi@unime.it (M.S.); 2Internal Medicine Unit, Department of Clinical and Experimental Medicine, University of Messina, 98122 Messina, Italy; veronica.nassisi@gmail.com (V.N.); sinicropidavide@gmail.com (D.S.); mariacinquegrani@gmail.com (M.C.); valentinacairo@libero.it (V.C.); gsquadrito@unime.it (G.S.); 3Medicine and Hepatology Unit, Department of Clinical and Experimental Medicine, University of Messina, 98122 Messina, Italy; stella91f@libero.it (M.S.F.); carlo.saitta@unime.it (C.S.); raimondo@unime.it (G.R.); 4Italian College of General Practitioners and Primary Care Professionals (SIMG), Section Messina, 98122 Messina, Italy; riccardoscoglio@gmail.com (R.S.); steniomarino@gmail.com (S.M.); alessandra-belvedere@virgilio.it (A.B.); 5Unit of Rheumatology, Department of Medicine, ARNAS Garibaldi Hospital, 95100 Catania, Italy; albertologullo@virgilio.it

**Keywords:** NAFLD, FCH, carotid atherosclerosis, noninvasive diagnosis, liver ultrasound, liver fibrosis, HOMA-IR

## Abstract

Familial combined hyperlipidemia (FCH) is a very common inherited lipid disorder, characterized by a high risk of developing cardiovascular (CV) disease and metabolic complications, including insulin resistance (IR) and type 2 diabetes mellitus (T2DM). The prevalence of non-alcoholic fatty liver disease (NAFLD) is increased in FCH patients, especially in those with IR or T2DM. However, it is unknown how precociously metabolic and cardiovascular complications appear in FCH patients. We aimed to evaluate the prevalence of NAFLD and to assess CV risk in newly diagnosed insulin-sensitive FCH patients. From a database including 16,504 patients, 110 insulin-sensitive FCH patients were selected by general practitioners and referred to the Lipid Center. Lipid profile, fasting plasma glucose and insulin were determined by standard methods. Based on the results of the hospital screening, 96 patients were finally included (mean age 52.2 ± 9.8 years; 44 males, 52 females). All participants underwent carotid ultrasound to assess carotid intima media thickness (cIMT), presence or absence of plaque, and pulse wave velocity (PWV). Liver steatosis was assessed by both hepatic steatosis index (HSI) and abdomen ultrasound (US). Liver fibrosis was non-invasively assessed by transient elastography (TE) and by fibrosis 4 score (FIB-4) index. Carotid plaque was found in 44 out of 96 (45.8%) patients, liver steatosis was found in 68 out of 96 (70.8%) and in 41 out of 96 (42.7%) patients by US examination and HSI, respectively. Overall, 72 subjects (75%) were diagnosed with steatosis by either ultrasound or HSI, while 24 (25%) had steatosis excluded (steatosis excluded by both US and HSI). Patients with liver steatosis had a significantly higher body mass index (BMI) compared to those without (*p* < 0.05). Steatosis correlated with fasting insulin (*p* < 0.05), liver stiffness (*p* < 0.05), BMI (*p* < 0.001), and inversely with high-density lipoprotein cholesterol (*p* < 0.05). Fibrosis assessed by TE was significantly associated with BMI (*p* < 0.001) and cIMT (*p* < 0.05); fibrosis assessed by FIB-4 was significantly associated with sex (*p* < 0.05), cIMT (*p* < 0.05), and atherosclerotic plaque (*p* < 0.05). The presence of any grade of liver fibrosis was significantly associated with atherosclerotic plaque in the multivariable model, independent of alcohol habit, sex, HSI score, and liver stiffness by TE (OR 6.863, *p* < 0.001). In our cohort of newly diagnosed, untreated, insulin-sensitive FCH patients we found a high prevalence of liver steatosis. Indeed, the risk of atherosclerotic plaque was significantly increased in patients with liver fibrosis, suggesting a possible connection between liver disease and CV damage in dyslipidemic patients beyond the insulin resistance hypothesis.

## 1. Introduction

Familial combined hyperlipidemia (FCH) is likely the most common inherited lipid disorder, since its estimated prevalence is 0.5–4% [1]. Although FCH was described already in 1973 [2], half century of progress has not led to a consensus on a simple and definitive set of diagnostic criteria [3,4]. FCH is characterized by a high risk of developing cardiovascular (CV) disease, and more specifically atherosclerotic cardiovascular disease (ASCVD), as well as metabolic complications including insulin resistance (IR) and type 2 diabetes mellitus (T2DM) [1,5,6]. The prevalence of non-alcoholic fatty liver disease (NAFLD) is widely acknowledged as significantly increased in FCH patients, especially if they also have IR or T2DM. There is increasing evidence that NAFLD may be associated with extrahepatic disorders especially in overweight or obese youths [7,8]. However, it is unknown how precociously metabolic and cardiovascular complications appear in FCH patients. NAFLD has already been suggested as an integral feature of FCH due to its wide overlap with combined hyperlipemic phenotype [6]. Furthermore, hepatic VLDL overproduction is driven by IR and hepatic fat accumulation [9], which establishes a vicious circle leading to the onset of overt NAFLD and T2DM. However, more recent evidence suggested that plasma insulin levels could be independently associated with the later onset of T2DM even after correction for sex, age, BMI, waist circumference, and FCHL status. For these reasons, in the current knowledge, high plasma insulin and IR status seem to be the drivers for the onset of T2DM and NAFLD in FCH patients [10]. On the other hand, IR has been associated with higher liver stiffness measurements (LSM) as evaluated by Transient Elastography (TE) in FCH patients, with a significantly higher prevalence in metabolic syndrome patients [11]. Consistent with the above-mentioned results, FCH patients with advanced fatty liver disease have higher LSM and are more likely to have high homeostasis model assessment index (HOMA-IR) or IR status [11].

The prevalence of NAFLD is about 10–24% in the general population [12], but it dramatically increases in FCH patients who have already experienced ASCVD, even without metabolic comorbidities, as reported in a small sample-sized study in 2004. In the same way, non-alcoholic steatohepatitis (NASH) has been associated with a higher incidence of coronary artery disease (CAD) than in patients with NAFL [13]. Indeed, NAFLD has been described in about the 76% of secondary prevention, metabolically healthy, FCH patients [14]. NAFLD diagnosis could be suspected by means of non-invasive scores, including the fatty liver index (FLI), the lipid accumulation product (LAP) index, the hepatic steatosis index (HSI), and the non-alcoholic fatty liver disease in metabolic syndrome patients scoring system (NAFLD-MS); all these score systems showed a good agreement with ultrasound diagnosis (US), but all the non-invasive tools are considered to be very far from the diagnostic performance offered by magnetic resonance-based methods [15,16].

Furthermore, carotid atherosclerosis risk is increased in FCH patients. Brouwers and coll. in 2009 estimated a 2.5-fold increase in plaque prevalence in FCH patients with respect to unaffected controls, as was the prevalence of increased arterial stiffness; however, FCH patients also presented with a 2.5-fold increased prevalence of metabolic syndrome in that study [17]. 

Carotid atherosclerosis and fatty liver disease are, thus, increased in FCH patients, and IR was suggested as the potential common link. 

The aim of this cross-sectional study was to assess the prevalence of carotid atherosclerosis, arterial stiffness, and fatty liver disease in a cohort of insulin-sensitive, non-obese, untreated FCH patients.

## 2. Materials and Methods

### 2.1. Patients Included in the Study

The study design consisted of several consecutive steps. In the pre-screening phase, 16,504 patients were evaluated by thirteen general practitioners from Messina and the surrounding area for possible enrollment in the study. Patients were selected in this first step if newly diagnosed for FCH, defined by apolipoprotein B (APOB) > 120 mg/dl, familial history of dyslipidemia or coronary atherosclerotic disease, and one of the following additional features: total cholesterol > 6.4 mmol/L, triglycerides > 1.5 mmol/L, high-density lipoprotein cholesterol (HDL-C) < 1 mmol/l (male) or < 1.3 mmol/L (female). Exclusion criteria were as follows: current treatment with any lipid lowering drug; T2DM; body mass index (BMI) > 30 kg/m^2^; any chronic disease (e.g., arterial hypertension, dysthyroidism, renal dysfunction, chronic inflammatory disease); current or past history of alcohol intake ≥ 2 UA/die; viral chronic hepatitis, autoimmune or genetic liver diseases; human immunodeficiency virus (HIV) infection; malignancies; and familial hypercholesterolemia (according to Dutch Lipid Clinic Network score [18]). Moreover, we chose to restrict the investigation to subjects aged 18–70 years.

A total of 144 patients met the aforementioned criteria and were screened at the Lipid Center of the Internal Medicine Unit (Messina University Hospital), where blood samples were repeated to determine lipid profile, fasting glucose and insulin to assess insulin sensitivity (by HOMA-IR [19]), thyroid function (determined by plasma thyrotropin [TSH] levels), and routine clinical chemistry. At this step, we chose stringent exclusion criteria as HOMA-IR > 2.5 or TSH > 2.5 mUI/L. Thus, 14 patients were subsequently excluded. Out of 130 selected subjects, 26 patients did not agree to continue the diagnostic investigations in the hospital specialist center. The remaining 104 subjects were referred by general practitioners to the Lipid Center from November 2018 to November 2020 (the hospital evaluation was stopped from April to September 2020 due to SARS-CoV-2 restrictions). Eight patients did not meet inclusion criteria after the first visit, and they were subsequently excluded. Finally, 96 patients (mean age 52.2 ± 9.8 years, 44 males, 52 females) were included in the study and underwent further clinical evaluation and instrumental assessments (carotid and abdomen ultrasound, non-invasive assessment of liver steatosis and fibrosis) (Figure 1). Written informed consent was obtained from all subjects in accordance with the Helsinki declaration. The study was approved by the Ethics Committee of the University of Messina (protocol number 2018/71).

### 2.2. Clinical and Laboratory Assessment

Demographic, clinical, and anthropometric data were collected at each step of the study. A 12 h overnight fasting blood sample was repeated at the time of enrollment to determine serum levels of alanine aminotransferase (ALT), aspartate aminotransferase (AST), gamma-glutamyl-transpeptidase (GGT), platelet (PLT) count, total cholesterol, HDL-cholesterol, triglycerides, APOB, blood glucose, glycated hemoglobin (HbA1c), and plasma insulin levels. The low-density lipoprotein cholesterol (LDL-C) was calculated according to the Friedewald formula, as follows: LDL-C = CT − (HDL-C + triglycerides/5). Insulin resistance was assessed with the HOMA method using the following equation: IR = [Fasting insulin (μU/mL) × Fasting glucose (mmol/L)]/22.5. BMI was calculated on the basis of the weight in kilograms and the height in meters. 

### 2.3. Carotid Artery Evaluation

Carotid ultrasound integrated with pulse wave velocity (PWV) assessment was performed twice by two experienced operators (DS and MC), separately, and validated by the supervisor (GM), using high-resolution B-mode ultrasonography (EsaoteMyLab 70 Gold, equipped with a linear array transducer 7–15 MHz). Carotid arteries were investigated in transverse and longitudinal projections of both the left and right side at the level of the common carotid artery, of the bulb, of the internal and external carotids. Carotid plaque was defined as a focal thickening of the arterial wall greater than 1.5 mm or exceeding 50% of the adjacent carotid intima media thickness (IMT), with lumen encumbrance and at least 0.5 mm of length [20,21]; moreover, as previously reported, IMT was estimated as the average of a six-points measurement, and carotid thickening was defined as an IMT ≥ 0.9 mm. Carotid–femoral pulse wave velocity (cf-PWV) was measured non-invasively by acquiring electrocardiogram gated waveforms at the carotid and femoral arterial sites [22].

### 2.4. Non-Invasive Assessment of Liver Steatosis and Fibrosis

Liver steatosis was diagnosed by both abdomen ultrasonography (US) and Hepatic Steatosis Index (HSI). The US assessment was performed in the morning, on the same day as fasting blood sampling and carotid US integrated with PWV, by one experienced operator (GS), using EsaoteMyLab X9 equipped with a convex 3.5 MHz probe. Presence of liver steatosis was defined through detection of parenchymal brightness, liver-to-kidney contrast, deep beam attenuation, bright vessel walls, and gallbladder wall definition [23]. Mild steatosis was recognized by a slight increase in liver echogenicity, a slight exaggeration of liver and kidney echo discrepancy and relative preservation of echoes from the walls of the portal vein. Moderate steatosis was recognized by loss of echoes from the walls of the portal vein, and a greater posterior beam attenuation, as well as discrepancy between hepatic and renal echoes. Finally, severe steatosis was recognized by a greater reduction in beam penetration, loss of echoes from most of the portal vein wall including the main branches, and a large discrepancy between hepatic and renal echoes. The HSI was calculated using the following formula: HSI = 8 × ALT/AST + BMI (+2 if T2DM, +2 if female). An HSI value higher than 36 was diagnostic for liver steatosis, whereas a value lower than 30 ruled out steatosis [24]. Liver fibrosis was investigated by both Fibrosis-4 (FIB-4) score and TE, on the same day of the other assessments. The FIB-4 score was calculated using the following formula: age (years) × AST (U/L)/(platelet 10^9^/L × √ALT (U/L). A cut-off lower than 1.30 excluded significant fibrosis (F2), and a cut-off higher than 2.67 predicted advanced fibrosis (F3) [25,26,27]. Liver TE was performed by one experienced operator (VN) and validated by the supervisor (GS), with the FibroScan^®^ (Echosense, Paris, France) medical device, using the M probe (called the standard probe). Results were accepted when at least 10 successful acquisitions with a success rate of at least 60% and an interquartile range (IQR) ≤30% were obtained [27]. A lower cut-off of 7.9 KPa excluded significant fibrosis (F2), and an upper cut-off of 9.9 KPa confirmed advanced fibrosis (F3) [28,29].

### 2.5. Statistical Analysis

According to available data on dyslipidemia in NAFLD patients, a power simple size of 100 patients ± 20% has been calculated. The Kolmogorov–Smirnov test was used to verify variables distribution. Numerical data were expressed as median and IQR, and categorical variables as number and percentage. Since some variables showed a non-normal distribution, a conventional non-parametric statistical approach was applied. According to the presence or absence of steatosis we identified two groups, and the Mann–Whitney test was used to compare the variables between the groups. Chi-squared testing was used to verify the interrelationships between categorical variables. Spearman’s test was used to verify the relationships between the variables. According to presence of steatosis, the Odds Ratio (OR) was estimated to assess the risk of presenting with a carotid plaque. Finally, a conditional backward stepwise logistic analysis model was designed to evaluate the risk of carotid atherosclerotic disease development. Statistical analysis was performed using SPSS software version 26. 

## 3. Results

The demographic, biochemical, and clinical characteristics of the 96 patients with newly diagnosed FCH are shown in Table 1. They were predominantly female (54.2%). Smoking habit was defined as active and past in 36 (37.5%) and 12 (12.5%) patients, respectively. Most of the patients declared they were non-habitual drinkers, while 44 subjects (45.8%) admitted a current alcohol intake <2 UA per day.

The carotid ultrasound evaluation detected at least one atherosclerotic plaque in 44 out of 96 patients (45.8%); abnormal cIMT was reported in 65 patients (67.7%). The non-invasive determinations of cf-PWV showed a median of 9.73 m/s with an IQR of 4.25. 

Liver steatosis was diagnosed in 64 out of 96 patients (66.7%) of the study cohort according to US. In detail, 39 patients (40.6%) had mild steatosis, 20 moderate (20.8%), and 5 severe (5.2%). Steatosis was absent in 32 patients (33.3%).

According to HSI evaluation, liver steatosis was diagnosed in 41 out of 96 patients (42.7%), whereas 55 patients (57.3%) presented with an HSI non-significant for steatosis; in detail, 11 subjects (11.5%) showed an HSI indicative of absence of steatosis, and 44 (45.8%) had an inconclusive HSI score. Two out of the 41 patients (4.9%) with an HSI score diagnostic for steatosis did not have signs of steatosis on ultrasonography evaluation. The remaining 39 patients (95.1%) showed the following grades of steatosis at liver US evaluation: 18 (43.9%) mild, 16 (39%) moderate, and five (12.2%) severe steatosis. Liver steatosis was confirmed by US in 22 out of the 44 patients (50%) with an inconclusive HSI score. In particular, 18 patients (40.9%) presented with a mild liver steatosis, four with moderate (9.1%), nobody with severe, whereas 22 patients (50%) did not show steatosis on US (Figure 2). When considering the 11 patients with an HSI score suggesting an absence of steatosis, in eight cases (72.7%) it was confirmed by US, whereas in three cases (27.3%) US identified a mild steatosis. 

On the other hand, 32 out of 96 patients (33.3%) had no ultrasonographic signs of steatosis, but this was confirmed in eight cases (25%) by an HSI indicative for absence of steatosis; two of these 32 patients (6.3%) showed an HSI score diagnostic for steatosis and 22 (68.8%) an inconclusive HSI score. Considering the 64 patients with US signs of steatosis, HSI score confirmed a diagnosis of steatosis in 39 cases (60.9%); three patients (4.7%) showed an HSI negative for steatosis diagnosis, and 22 (34.4%) an inconclusive score.

A little discrepancy between HSI and US when HSI was indeterminate was noted; however, when HSI is used to rule out NAFLD, a significant concordance with US steatosis was confirmed (Cohen’s kappa 0.705). 

Overall, 66 (68.8%) subjects were diagnosed with steatosis by either ultrasound or HSI, while the group without steatosis (both by US and HSI) comprised 30 (31.3%) cases (Table 2).

Liver steatosis had a statistically significant correlation with fasting insulin (*p* < 0.05), BMI (*p* < 0.05), and an inverse correlation with HDL-cholesterol (*p* < 0.05). 

According to TE evaluations, a significant liver fibrosis (>7.9 KPa) was found in eight patients (8.3%), one of whom had an LSM suggestive of advanced fibrosis (LSM > 9.9 KPa). Eighty-eight (91.7%) patients showed absence or mild fibrosis (LSM < 7.9 KPa). 

According to the FIB-4 index, 71 patients (73.9%) had no fibrosis, while 25 (26.1%) patients had undetermined results; no patients reached a score >2.67 (Table 1). 

Liver fibrosis assessed by TE was significantly associated with BMI (*p* < 0.001) and cIMT (*p* < 0.05). On the other hand, liver fibrosis assessed by the FIB-4 index had statistically significant correlation with sex (*p* < 0.05), cIMT (*p* < 0.05), and atherosclerotic plaque (*p* < 0.05). 

When we tested the potential association between any grade of liver fibrosis assessed by the FIB-4 index and the risk of presenting with atherosclerotic plaque or abnormal cIMT, we found a statistical significance in the first case (OR 6.667, 95% Confidence Interval [CI] 2.213–20.087, *p* < 0.001), but not in the second (OR 1.689, 95% CI 0.553–5.161, *p* = 0.358). When liver fibrosis was identified by TE, we found a more than 4-fold increased risk of presenting with atherosclerotic plaque, although it did not reach the statistical significance (OR 4.778, 95% CI 0.954–23.938, *p* = 0.057), as well as abnormal cIMT risk (OR 3.696, 95% CI 0.443–30.865, *p* = 0.227) (Table 3).

According to the estimation of liver fibrosis, a stepwise multivariable logistic regression model was designed to evaluate the risk of presenting with atherosclerotic disease. Consistent with the pathophysiology, we tested the dependence of the variable “plaque presence” from the FIB-4 index, corrected for smoking habit, alcohol consumption, and sex, as well as HSI and TE. The regression model showed that the FIB-4 score was an independent significant predictor of plaque (OR 6.863, *p* < 0.001) and smoke habit (OR 2.122, *p* = 0.047). Alcohol consumption, sex, HSI score, and liver stiffness measured by TE were not significantly associated with the presence of plaque and were removed from the final step of the analysis (Table 4). 

The FIB-4 score maintained its ability to predict vascular damage (assessed by cIMT) also when they were considered as continuous variables (beta coefficient 0.413, *p* < 0.001); in detail, FIB-4 was found to be a potential predictor of cIMT over HSI score and liver stiffness as measured by TE.

Furthermore, a multivariable logistic regression model was estimated to find potential predictors of arterial stiffness (PWV) among the independent variables already included in the model(s) for plaque/cIMT; among liver steatosis (assessed by HSI) and fibrosis scores assessed by both FIB-4 and TE, we estimated no significant models.

On the other hand, plasma insulin levels were not associated to PWV values (beta coefficient–0.051, *p* = 0.795).

## 4. Discussion

The close interconnection between the main contributors to cardiometabolic disease in FCH patients has been widely investigated so far. T2DM is significantly more prevalent in FCH patients than in non-FCH patients. In 2020, Brouwers and coll. published the results of their survey with a 15-year follow-up in an FCH cohort, in which the risk of developing T2DM showed a six-fold increase over time in FCH patients as compared to their spouses; notably, the risk of unaffected relatives developing T2DM without the diagnostic criteria for FCH was the same as in index cases, suggesting that genetic background plays an important role, maybe also on the dietary and physical behavior [5]. NAFLD has already been suggested as a common feature among T2DM and FCH [6], but also, in the absence of overt T2DM, IR status has been acknowledged to play an important role. In fact, hepatic VLDL overproduction driven by IR and hepatic fat accumulation establish a vicious circle leading to the onset of NAFLD and T2DM on an FCH background [9,10]. Moreover, in FCH subjects as well as in their apparently unaffected relatives or family members, obesity was an important contributor to the grade of fatty liver [6].

This link between NAFLD and dyslipidemia has been deeply investigated in the last 20 years [30,31,32]; in fact, it has been reported that 69% of NAFLD patients and 72% of those with NASH have dyslipidemia or hyperlipidemia [33,34]. Similarly, the prevalence of NAFLD has been investigated in high risk groups [35], ranging between 60 and 70% in the Italian population to 42.6% in the UK population with T2DM [36,37,38,39], 78.8% among patients with metabolic syndrome [37], and 50% in patients with dyslipidemia [40]. Moreover, in a large cross-sectional study on 44,767 patients, the overall prevalence of NAFLD was 53%, whereas in patients with high TC/HDL-C and TG/HDL-C ratios, the estimated prevalence was 78% [35,41]. In fact, the growing body of evidence suggests that cardiovascular disease in patients with NAFLD determines the outcome rather than liver disease progression [42].

The other corner stone of what we currently know about the FCH patient is represented by the high CV risk. In fact, FCH patients develop atherosclerotic lesions prematurely and experience ASCVD precociously [17]. The CV risk in FCH patients increases over time with advancing age, but it is often attributed to the risk enhancers commonly complicating this setting. In particular, IR, T2DM, overweight, increased waist circumference, and arterial hypertension are very frequently considered in arterial and metabolic worsening in FCH patients [1,4,43]. It has been reported that carotid atherosclerosis risk affects FCH subjects up to 2.5-fold more than unaffected controls, as well as subclinical atherosclerosis assessed by arterial stiffness estimation. However, this evaluation has often been carried out among FCH patients already presenting with cardiometabolic complications, including IR or metabolic syndrome [17].

Clinical or subclinical carotid atherosclerosis and fatty liver disease generally coexist in FCH patients, and IR has been already suggested as the potential common link [1,5,6,9,10]. However, the pathophysiological relationship between IR, NAFLD, and arterial wall thickening should be further investigated, even if the genetic variants consistently associated with NAFL are not associated with measures of sub-clinical atherosclerosis [44].

The potential interrelationship between carotid IMT and LSM (assessed by TE) was not statistically supported in our previous study, where primary prevention, NAFL-diseased, insulin-resistant FCH patients were enrolled and evaluated [11]. It has already been suggested that vascular damage may also advance in FCH when a statin lipid lowering therapy is prescribed, but considering the lipid target suggested at the time of the study, we cannot state whether, according to the current further lowered lipid targets, the results will remain the same [45,46].

It is important to underline that the high CV risk of FCH patients should be not disregarded when they are not complicated with T2DM/IR or arterial hypertension [46] However, the FCH condition has been poorly considered beyond the potential contribution of these several co-players. The FCH subject is definitely a patient who represents the current concept of “cardiometabolic risk”.

Within our cohort of insulin-sensitive, non-obese, normotensive, untreated FCH patients, we found a large number of subjects already affected by liver steatosis, as identified by either US or HSI (75%). Notably, this prevalence is similar to that detected in subjects with metabolic syndrome and in those with high TC/HDL-C or TG/HDL-C ratios [37,41].

Unfortunately, neither HSI nor US can be considered gold standard for diagnosis and grading of hepatic steatosis, but they are inexpensive and non-invasive, and HSI could be easily included in selected patients as a useful screening tool in any healthcare setting [15,47].

According to TE, the prevalence of fibrosis in our study population was 8.3%, with only one patient with advanced fibrosis. This finding can be partly explained by the relatively young age of the population studied (median age: 53 years). Indeed, we enrolled people at first diagnosis of dyslipidemia, thus excluding the possible confounding factor of lipid lowering drugs in evaluating either the prevalence or severity of steatosis and fibrosis.

We did not find a significant ability of liver steatosis (assessed by HSI or US) and fibrosis (assessed by FIB4 or TE) to predict PWV values in our population. Nevertheless, a FIB-4 score higher than 1.30 predicted the risk of presenting with atherosclerotic plaque already at FCH diagnosis (OR: 4.624), confirming this statistical significance also after correction for smoking habit, sex, and alcohol consumption. Thus, the FIB-4 score appeared to be able to predict atherosclerotic disease in FCH patients independently and over HSI score and hepatic fibrosis evaluated by TE. Additionally, the FIB-4 score confirmed its ability to predict cIMT measures also when they were considered as continuous variables, again over HSI score and liver stiffness measured by TE.

NAFLD patients commonly show a dyslipidemia profile characterized by high triglycerides, low HDL-C, and high LDL-C, features that are the same for our cohort of patients. It has been hypothesized that dyslipidemia in NAFLD is caused by the same pathogenic mechanisms that also drive hepatic steatosis, including insulin resistance [48]. Furthermore, the close association between NAFLD and atherosclerosis is well established, and NAFLD is nowadays considered a risk factor for atherosclerosis [49], even if this risk may differ between metabolically- vs. genetically-driven steatosis [50].

Few studies have investigated the association between FIB-4 and atherosclerosis. In a recent study, Xin and colleagues demonstrated an association between the progression of atherosclerosis detected by brachial ankle pulse wave velocity (ba-PWV) and higher incidence of NAFLD, with high risk of liver fibrosis as assessed by NAFLD fibrosis score and FIB-4, whereas they found no association between cIMT and fibrosis [51]; however, the insulin-sensitivity status was not assessed in that study. The PWV predicted incident NAFLD as well as liver fibrosis (non-invasively assessed by specific scores) in a cohort of patients not only affected by FCH, but also by arterial hypertension, obesity, T2DM, and dyslipidemia.

Histologically significant fibrosis (F2-F4) was found to be associated with higher ba-PWV [52]; older age, arterial hypertension, and advanced liver fibrosis, even when assessed by FIB-4 and NAFLD fibrosis score, were confirmed as risk factors for atherosclerosis in NAFLD patients. It has been already demonstrated that FIB-4 can predict the risk of coronary artery calcification in NAFLD patients [53,54]. However, none of these studies considered the presence or absence of plaque. In our study, FIB-4 predicted the presence of atherosclerotic plaque already at the diagnosis. Insulin resistance is a well acknowledged risk factor for both endothelial and arterial damage progression and liver steatosis. Thus, we excluded insulin-resistant patients from the study design; the association between FIB-4 and atherosclerotic plaque could suggest further mechanisms linking fat ectopic accumulation both in the liver and in arterial walls.

One of the main strengths of this study could be represented by the restrictive inclusion/exclusion criteria; in fact, we selected a study population as homogeneous as possible, avoiding the potential effect of overtly overweight, insulin resistance, T2DM, thyroid disease, or arterial hypertension. By selecting only non-diabetic, normotensive, non-obese subjects with “normal” HOMA-IR (≤2.5), we could expect to find no association between liver fibrosis indices and fasting insulin or HOMA or HbA1c. Indeed, we found a significant association between steatosis and fasting insulin, even with insulin plasma levels within normal range. Moreover, the ability of FIB-4 to predict the presence of plaque in this specific setting of FCH patients appears extremely interesting and promising.

Of course, our study has limitations. In fact—consistent with its design—this study cannot enable us to derive absolute and actual data for the prevalence of insulin-sensitive FCH subjects among the general population, nor for clinical or subclinical carotid atherosclerosis among insulin-sensitive FCH, or for NAFLD and hepatic fibrosis in this clinical setting. In addition, it should be noted that both liver steatosis and fibrosis were evaluated using non-invasive approaches.

Although robust statistically significance is lacking-also due to the minimum requested simple size-FIB-4 can be suggested as an easy screening tool to assess liver fibrosis, but, in particular, as an adjunctive tool in clinical practice to better stratify dyslipidemic or NAFLD patients at risk of cardiovascular and liver disease progression.

## 5. Conclusions

In conclusion, according to the results obtained in this study, we propose to screen for NAFLD all patients with “pure” (or “isolated”) primary combined dyslipidemia, namely all subjects with a suspected FCH lipid profile but with no comorbidities or other metabolic risk factors. In this setting, the HSI could be effectively used as a first screening method (due to its non-invasiveness as a tool and easily calculable score), also by general practitioners and first- and second-level ambulatory physicians, who can, thus, decide the best candidate patients for further checks at a third level specialist center. In fact, our data show a 75% prevalence of hepatic steatosis in these selected patients, with a significantly increased risk of atherosclerotic disease among those with liver fibrosis. FIB-4 can also be suggested as an easy screening tool to assess liver fibrosis, especially as an adjunctive tool to better stratify dyslipidemic or NAFLD patients at risk of cardiovascular and liver disease progression.

These data are of particular interest because they further emphasize the strong link between altered lipid metabolism and liver disease. They also make it clear that patients with suspected FCH should be investigated for atherosclerosis and liver disease already at diagnosis, in order to promptly detect patients at higher risk of CV or liver disease progression.

## Figures and Tables

**Figure 1 biomedicines-10-01770-f001:**
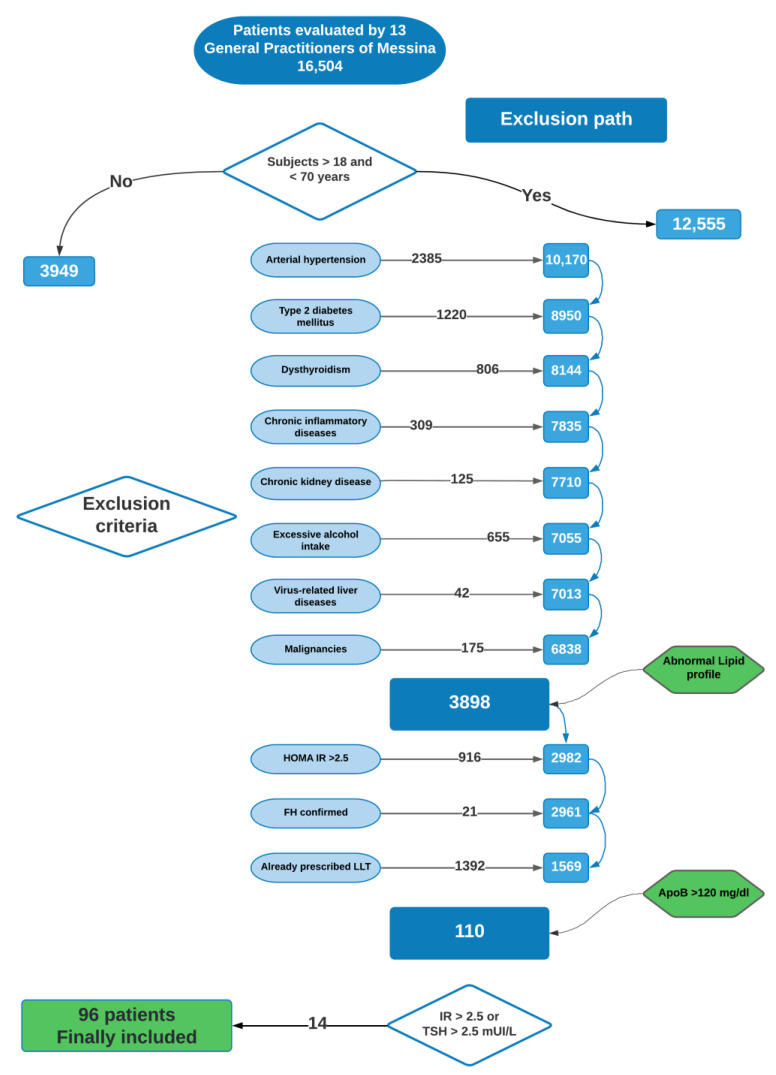
Flow diagram for inclusion/exclusion path. (Flow diagram was drawn by Lucidchart©, Lucid Software Inc., 2022, South Jordan, UT; www.lucidchart.com; accessed date: 17 June 2022).

**Figure 2 biomedicines-10-01770-f002:**
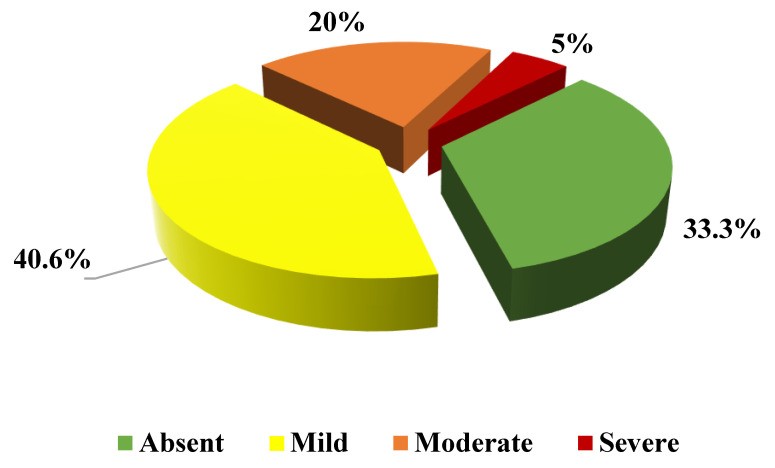
Evaluation and grade of liver steatosis performed by abdomen ultrasound in 44 FCH patients with indeterminate HSI (30 < HSI < 36).

**Table 1 biomedicines-10-01770-t001:** Demographic, clinical and laboratory characteristics of 96 insulin-sensitive FCH patients included in the study.

**Male sex**, n [%]	44 (45.8)
**Age**, years	53 (12)
**BMI**, Kg/m^2^	25.5 (4.9)
**Total cholesterol**, mg/dL	246.5 (26)
**HDL-C**, mg/dL	50.5 (21)
**Triglycerides**, mg/dL	119 (85)
**LDL-C**, mg/dL	169 (21)
**Fasting glucose**, mg/dL	89 (11)
**HbA1c**, %	5.4 (0.6)
**Fasting insulin**, mIU/L	7.1 (3.6)
**HOMA-IR**	1.6 (0.8)
**AST**, IU/L	20 (10)
**ALT**, IU/L	20 (15)
**GGT**, IU/L	24 (26)
**PLT count** × 10^3^/mmc	240 (85)
**Apolipoprotein A**, mg/dL	154.5 (49)
**Apolipoprotein B**, mg/dL	131 (26)

Data are shown as median and IQR or as number and percentage (%). Abbreviations: BMI, body mass index; HDL-C, high-density lipoprotein cholesterol; LDL-C, low-density lipoprotein cholesterol; HbA1c, glycated hemoglobin; HOMA, homeostasis model assessment; AST, aspartate aminotransferase; ALT, alanine aminotransferase; GGT, gamma-glutamyl-transpeptidase; PLT, platelet.

**Table 2 biomedicines-10-01770-t002:** Noninvasive and ultrasonographic characteristics of 96 insulin-sensitive FCH patients included in the study.

**FIB-4**	0.9 (0.7)
**HSI**	34.9 (6.4)
**LSM**, KPa	5.1 (2.2)
**cIMT**, mm	1 (0.30)
**PWV**, m/s	9.7 (4.3)
**Steatosis according to liver US**, n [%]	
Absent	32 (33.3)
Mild	39 (40.6)
Moderate	20 (20.8)
Severe	5 (5.2)
**Steatosis according to HSI evaluation**, n [%]	
HSI < 30	11 (11.5)
30 < HSI < 36	44 (45.8)
HSI > 36	41 (42.7)
**Liver fibrosis according to LSM by TE**, n [%]	
LSM < 7.9 KPa	88 (91.7)
7.9 KPa > LSM < 9.9 KPa	7 (7.3)
LSM > 9.9 KPa	1 (1)
**Liver fibrosis according to FIB4**, n [%]	
FIB4 < 1.30	71 (74)
1.45 < FIB4 < 2.67	25 (26)
FIB4 > 2.67	0 (0)

Abbreviations: FIB-4, fibrosis-4 score; HSI, hepatic steatosis index; LSM, Liver stiffness measurements; TE, transient elastography. Data are shown as median and IQR or as number and percentage (%).

**Table 3 biomedicines-10-01770-t003:** Potential association between any grade of liver fibrosis (assessed by FIB-4 or TE) and the risk of presenting with atherosclerotic plaque or abnormal cIMT.

	Atherosclerotic Plaque			Abnormal cIMT		
	**OR**	**95% CI**	** *p* **	**OR**	**95% CI**	** *p* **
**Fibrosis by FIB-4**	6.667	2.213–20.087	**<0.001**	1.689	0.553–5.161	0.358
**Fibrosis by TE**	4.778	0.954–23.938	0.057	3.696	0.443–30.865	0.227

**Table 4 biomedicines-10-01770-t004:** Conditional backward stepwise logistic regression analysis of possible predictive factors of carotid plaque development in newly diagnosed FCH patients.

	Final Step of Multivariable Logistic Regression Analysis		
**Variables**	**OR (IC 95%)**	**IC 95%**	** *p* **
**FIB-4**	6.863	2.167–21.739	<0.001
**Smoking habit**	2.122	1.009–4.462	0.047
**LSM by TE**	4.031	0.731–22.237	0.110

## Data Availability

All data are available upon any reasonable request.

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
