# Peer review of "Fatty Liver as Potential Biomarker of Atherosclerotic Damage in Familial Combined Hyperlipidemia"

_biomedicines, 2022, doi:10.3390/biomedicines10081770_

Round 1
Reviewer 1 Report
The paper by Mandraffino and colleagues reports a study establishing a correlation between an increased cardiovascular risk and NAFLD, in newly diagnosed patients with familial combined hyperlipidemia, in the absence of comorbidities usually associated with an increased cardiovascular risk.
This research work goes beyond the "classic" link between NAFLD and atherosclerosis via a condition of IR or T2MD (patients with these two conditions are indeed excluded from this study).
The introduction of the work is clear, complete exhaustive and well written. The sample selection process was careful and scrupulous, the experimental part is well reported and the exams still validated by an independent expert supervisor.
The research results are new and interesting, and can be extremely useful in terms of a general prevention approach towards cardiovascular and metabolic risk. This work also provides a base for future studies (i.e. repeating the study in different cohorts or using diagnostic methods that enable a better sample stratification and interpretation of results).
On the basis of these consideration, I am convinced that the paper should be accepted for publication in Biomedicines. I would just suggest the authors to consider the following minor points:
1) Could the authors comment/explain the discrepancy (in some cases misleading) between the HSI and US test results highlighted for the reference sample?
2) Table 1 should be inserted in a single page for a more immediate and easy consultation, perhaps Figure 2 should be moved to another page.
Author Response
... following minor points:
First, we want to thank very much the referee for these very encouraging comments; please, find below our replies to the points you raised:
- Could the authors comment/explain the discrepancy (in some cases misleading) between the HSI and US test results highlighted for the reference sample?
- Thank you very much for this comment; indeed, HSI and US detected globally 75% of NAFLD subjects within our sample population; however, not both the tests always detected the same patients, or they assigned the same patient to an unambiguous grade of steatosis. This issue is well reported in literature. However, to explain this discrepancy, we performed an agreement test by Cohen’s kappa; we found an index of 0.464 in ruling out NAFLD between HSI and US, but when we censored the indeterminate values the kappa index was 0.702. The AUC of HSI as continuous value on US steatosis was of 0.817, very similar to those reported in literature. Moreover, the beta index for HSI in predicting US steatosis was 0.589 in a linear regression model (p<0.001). Unfortunately, both HSI and US are considered very far from being the gold standard test to detect hepatic steatosis, but they are definitely unexpensive tests, with HSI very cheap and practicable and low-cost clinical tool. We tried to add some informative sentence within the text reflecting this aspect as you can find it now in the revised manuscript.
- Table 1 should be inserted in a single page for a more immediate and easy consultation, perhaps Figure 2 should be moved to another page.
- Many thanks for this advice; indeed, the table 1 consisted of a number of rows, very difficult to adapt in one page as it is embedded in the text; so, we have divided the previous table 1 into 2 new tables, now 1 and 2, in order to allow a more effective paging. I hope that this could improve the ease of reading and consultation.

Reviewer 2 Report
Dear Authors my compliments for your research.
I have not further comments, only a minor detail, you forgot to insert key words in the abstract.
Author Response
Dear Authors my compliments for your research.
- I have not further comments, only a minor detail, you forgot to insert key words in the abstract.
- Dear Referee, thank you very much for your very encouraging comments; in the revised manuscript we added the key words in the title page.
